# Energy Metabolism in Mouse Sciatic Nerve A Fibres during Increased Energy Demand

**DOI:** 10.3390/metabo12060505

**Published:** 2022-05-31

**Authors:** Laura R. Rich, Bruce R. Ransom, Angus M. Brown

**Affiliations:** 1School of Life Sciences, University of Nottingham, Nottingham NG7 2UH, UK; mszlrr@exmail.nottingham.ac.uk; 2Department of Neurology, School of Medicine, University of Washington, Seattle, WA 98195, USA; bransom@uw.edu; 3Department of Neuroscience, City University of Hong Kong, Hong Kong

**Keywords:** glucose, lactate, glycogen, recovery

## Abstract

The ability of sciatic nerve A fibres to conduct action potentials relies on an adequate supply of energy substrate, usually glucose, to maintain necessary ion gradients. Under our ex vivo experimental conditions, the absence of exogenously applied glucose triggers Schwann cell glycogen metabolism to lactate, which is transported to axons to fuel metabolism, with loss of the compound action potential (CAP) signalling glycogen exhaustion. The CAP failure is accelerated if tissue energy demand is increased by high-frequency stimulation (HFS) or by blocking lactate uptake into axons using cinnemate (CIN). Imposing HFS caused CAP failure in nerves perfused with 10 mM glucose, but increasing glucose to 30 mM fully supported the CAP and promoted glycogen storage. A combination of glucose and lactate supported the CAP more fully than either substrate alone, indicating the nerve is capable of simultaneously metabolising each substrate. CAP loss resulting from exposure to glucose-free artificial cerebrospinal fluid (aCSF) could be fully reversed in the absence of glycogen by addition of glucose or lactate when minimally stimulated, but imposing HFS resulted in only partial CAP recovery. The delayed onset of CAP recovery coincided with the release of lactate by Schwann cells, suggesting that functional Schwann cells are a prerequisite for CAP recovery.

## 1. Introduction

The critical role played by astrocytic glycogen in learning and memory [1] has focused interest on the putative role(s) of central nervous system glycogen in physiological processes [2,3]. The presence of glycogen in peripheral nerve Schwann cells was recently established [4], introducing a wider role for nervous system glycogen. There are fundamental similarities in the role(s) of glycogen in peripheral and central nerves, which include a glial cell location and conversion to lactate for subsequent transport to axons [5,6], but also important differences. Whereas all axons are considered to benefit from the presence of glycogen in astrocytes in central white matter [5], only the A fibre axons in the sciatic nerve are supported by glycogen during periods of aglycaemia; the C fibres do not benefit from the presence of glycogen [4]. Emerging information from central axons indicates flexibility in substrate use depending upon the energy demand placed on the tissue [7,8], at the core of which is lactate transport from astrocytes to neurones. However, the source of interstitial lactate is not exclusively glycogen, since it can also be derived from fructose and glucose [9], requiring a more holistic appreciation of the conditions under which glycogen is metabolised to lactate.

There remains a distinct lack of information relating to peripheral nerve metabolism and substrate use. This has particular relevance to diabetes, where the sciatic nerve is exquisitely sensitive to the condition, with the first symptoms of the disease emanating from the peripheral nerve [10]. Whether substrate specificity is related to this sensitivity is unknown. The aim of this paper was to study conditions under which glycogen is converted to lactate to support the A fibre CAP, in nerves exposed to HFS, with the goal of enhancing our understanding of the important role Schwann cells play in peripheral nerve metabolism. We show that the metabolism of glycogen to lactate is dependent upon the prevailing glucose concentration, with supra-physiological concentrations of glucose sparing glycogen. Such versatility suggests flexibility in the sciatic nerve relating to fibre specific energy substrate use comparable to that of central nervous system (CNS) axons.

## 2. Results

### 2.1. Glycogen Supports the CAP in Substrate-Free aCSF

In order to test the manner in which glycogen supports the A fibre CAP, we exposed sciatic nerves to substrate-free aCSF, where the latency to failure of the CAP is an accurate indicator of glycogen depletion [4,5]. In nerves stimulated at 1 Hz, the CAP fell after about 100 min, a latency that was halved when 200 μM CIN, an inhibitor of lactate uptake into axons, was included in the aCSF (Figure 1A,B). Imposing high-frequency 100 Hz stimulus (HFS) in mouse optic nerve (MONs) perfused with substrate-free aCSF resulted in glycogen depletion and acceleration of CAP failure [5]. A qualitatively similar result was obtained in sciatic nerves. However, the combined effects of 200 μM CIN and HFS in substrate-free aCSF were not additive (Figure 1B). These data indicate that Schwann cell glycogen is metabolised to support CAP conduction during exposure to substrate-free aCSF, and that limiting the supply of glycogen-derived lactate to the A fibre axons by either depleting glycogen, imposing HFS, or blocking lactate uptake into axons with CIN, significantly accelerated CAP failure.

### 2.2. Increasing aCSF Glucose Restores the CAP

We investigated the effects of imposing HFS on the ability of the nerve to sustain CAP conduction, since imposing HFS in MONs perfused with 10 mM glucose aCSF resulted in glycogen depletion [5]. Sciatic nerves perfused with 10 mM glucose and stimulated at 1 Hz were able to fully support the CAP for 8 h. Increasing the stimulus frequency, however, prevented complete maintenance of the CAP, with a gradual decline in CAP amplitude seen after about 4 h (Figure 2A). Including 200 μM CIN in 10 mM glucose aCSF perfusing nerves exposed to HFS caused a rapid fall in the CAP amplitude (Figure 2A,B). Increasing the glucose to 30 mM during HFS fully supported the CAP for 8 h, and addition of CIN had no effect (Figure 2B). These data suggest that glycogen is depleted after 4 h in sciatic nerves exposed to 100 Hz stimulus, and thereafter, 10 mM glucose cannot fully maintain the CAP, which supports the notion that in 30 mM glucose glycogen is not depleted and axon uptake of glycogen-derived lactate is not required as a supplemental substrate.

### 2.3. Lactate Supports the CAP

Substituting 10 mM glucose with 20 mM lactate resulted in a small but significant fall in the CAP area (Figure 3A), whereas superfusing with a combination of 10 mM glucose and 20 mM lactate fully supported the CAP to a greater extent than either substrate individually (Figure 3B). These data indicate a degree of versatility with regard to the substrate used by the nerve, where the nerve can simultaneously use glucose and lactate.

### 2.4. Pre-Incubation with High-Frequency Stimulus Causes CAP Failure

Since imposing HFS on MONs perfused with 10 mM glucose caused glycogen depletion [5], we tested whether pre-incubation of the nerve in 10 mM glucose and HFS for increasing durations had any effect on the latency to CAP failure during subsequent exposure to substrate-free aCSF. With a control pre-incubation of 20 min in 10 mM glucose stimulated at 1 Hz, the latency to CAP failure on exposure to substrate-free aCSF was about 100 min. This latency fell significantly after pre-incubation for 2, 4, or 6 h in 10 mM glucose and HFS (Figure 4A,C). These data are indicative of significant glycogen depletion in the tissue, even in the presence of 10 mM glucose, when exposed to HFS, and that the glycogen depletion was complete after 2 h. We hypothesise, based on the data presented in Figure 2, that pre-incubation of nerves in 30 mM glucose exposed to HFS would not deplete glycogen (Figure 4B,D). Indeed, the data reveal that the latency to CAP failure actually increased after pre-incubation in 30 mM glucose (Figure 4A,D), suggesting hyperglycaemic pre-incubation treatment increases glycogen content.

### 2.5. CAP Rescue

The loss of the CAP after exposure to aglycaemic conditions affords the opportunity to test substrate efficacy in restoring the CAP in the absence of glycogen, since exposure to aglycaemia depletes glycogen [4]. In nerves perfused with 10 mM glucose aCSF then exposed to 1 Hz stimulus, substituting substrate-free aCSF caused CAP failure after about 100 min. When the CAP started to fail, addition of either 10 mM glucose, 20 mM lactate or 30 mM fructose was able to fully restore the CAP, although there was a significant delay between introducing the substrate and recovery of the CAP (Figure 5). These experiments were important in establishing that glycogen depletion and the subsequent failure of the CAP do not necessarily lead to axon death, and timely substrate re-introduction fully restores the CAP.

### 2.6. Substrate Restoration of the CAP during High-Frequency Stimulus

The ability of substrates to support the CAP was contrasted with the ability of the substrate to recover the CAP during HFS to determine if the increased energy burden would affect the CAP recovery. In these experiments, the nerve was stimulated at 1 Hz and exposed to substrate-free aCSF. When the CAP had fallen to 50% of its baseline amplitude, the substrate was introduced accompanied by HFS. The principal result was that 10 mM glucose, 20 mM lactate, and 30 mM glucose were able to partially restore the CAP, with 30 mM glucose resulting in the greatest recovery and 20 mM lactate the least (Figure 6A,D), which is to be expected from the results in Figure 3. There was no significant difference in the latency for CAP recovery among the different substrates (Figure 6C). An interesting finding was that in the presence of 10 mM glucose and CIN, the CAP was only temporarily rescued, as it began to fall after the initial recovery period, whereas with 30 mM glucose and CIN, the CAP recovery was sustained (Figure 6B).

### 2.7. Lactate Elevations Precede CAP Recovery

The potential role of Schwann cell function in restoring the CAP was investigated using lactate sensors. We have previously shown that when the CAP was recovered in 20 mM fructose the lactate signal increase preceded CAP recovery [9], an effect also evident with re-introduction of 10 mM glucose aCSF, where the lactate ([lac]_o_) increased rapidly before the CAP recovered (Figure 7A). This effect was also present in nerves exposed to 100 Hz stimulus and 10 mM glucose plus 200 μM CIN during the recovery period (Figure 7B).

## 3. Discussion

The results presented in this paper demonstrate two principle features of sciatic nerve A fibre conduction. (1) Increasing the tissue energy demand by imposing HFS leads to delayed loss of the CAP in nerves perfused with 10 mM glucose; imposing HFS tips the balance such that energy demand exceeds substrate supply. Increasing glucose to 30 mM fully maintains the CAP; the supply of energy substrate now exceeds demand. (2) The energy substrate that fuels sciatic nerve A fibre conduction can be glucose, glycogen-derived lactate, glucose-derived lactate, or exogenously applied lactate, which can be utilised simultaneously. In this manner, the sciatic nerve demonstrates a degree of metabolic versatility on a par with the MON, a central white matter tract.

### 3.1. CAP Failure Signals Depletion of Utilisable Glycogen

Some of the conclusions reached in this paper rely on the assumption that the sciatic nerve A fibre CAP fails when utilisable Schwann cell glycogen is exhausted [4]. The onset of CAP correlates with depletion of ‘utilisable’ glycogen and, when compared to glycogen assay, acts as a complementary monitor of glycogen exhaustion since (i) some biochemically measurable glycogen remains in sciatic nerves [4] and optic nerves [5] exposed to substrate-free aCSF, which is assumed to result from residual, but inert, glucosyl molecules attached to the glycogenin skeleton [11]; (ii) glycogen phosphorylase inhibitors prevent glycogen degradation even when glycogen content is high [12]; (iii) the glycogen content in MON at the onset of aglycaemia determines the latency to CAP failure, indicating the steady rate of glycogen metabolism under equivalent conditions [5]; and (iv) glycogen is metabolised to provide supplemental lactate in MONs perfused with hypoglycaemic glucose, where the CAP fails when the glycogen is exhausted [5].

High-frequency stimulus was used extensively in these experiments as a means of increasing tissue energy demand [5,13] in order to alter the balance between energy demand and substrate supply. Glycogen, located in Schwann cells in the sciatic nerve, is metabolised to lactate and transported to axons, where it supports the A fibre CAP during periods of glucose withdrawal [4]. Imposing extended periods of HFS, designed to increase tissue energy demand and deplete glycogen, or blocking lactate uptake into axons with CIN, accelerates CAP failure during glucose withdrawal (Figure 1). The CAP was fully supported for 8 h in nerves bathed in 10 mM glucose and stimulated at 1 Hz. When HFS was imposed, the CAP fell after 4 h and gradually declined (Figure 2A). This CAP failure likely coincides with the exhaustion of glycogen; for up to 4 h in 10 mM glucose, the energy demand placed on the tissue by HFS can be met by the combination of 10 mM glucose in the perfusate and the Schwann cell glycogen, but upon exhaustion of the glycogen, the CAP starts to fall, since 10 mM glucose is unable to fully support the CAP. This assumption is supported by the rapid failure of the CAP in nerves exposed to 10 mM glucose and HFS in the presence of CIN (Figure 2A) and the accelerated CAP failure in nerves exposed to aglycaemia after pre-incubation in 10 mM glucose with HFS (Figure 4A). However, pre-incubation for up to 4 h in 30 mM glucose with HFS results in elevated glycogen, indicated by the increased latency to CAP failure during a period of subsequent aglycaemia (Figure 4B); presumably, the glucose can both fully support the CAP and donate glucosyl units for storage as glycogen (Figure 4B). The full maintenance of the CAP when the aCSF glucose is increased to 30 mM with added CIN (Figure 2A), and the additive effect of 10 mM glucose and 20 mM lactate on the CAP amplitude (Figure 3A) indicates versatility in axonal use of energy substrate. High concentrations of glucose alone, or a combination of glucose and lactate (Figure 3B), as likely occur when glycogen/glucose is shuttled via the Schwann cell as lactate, fully support the CAP.

### 3.2. CAP Restoration

The loss of the CAP when stimulated at 1 Hz, resulting from exposure to substrate-free aCSF, could be reversed and the CAP fully restored if 10 mM glucose, 20 mM lactate, or 30 mM fructose was introduced. Loss of the CAP does not inevitably lead to axon damage, providing the substrate is re-introduced in a timely manner, and CAP rescue does not require glycogen, whose exhaustion precedes CAP failure (Figure 5). Exposing nerves to substrate-free aCSF, then imposing HFS when the CAP had fallen to 50% of its baseline amplitude, along with addition of substrate, showed a qualitatively similar pattern, with incomplete recovery (about 80%) of the CAP in all conditions (Figure 6), showing the metabolic burden of HFS caused a significant number of axons to fail to recover. An intriguing observation was the temporary restoration of the CAP in 10 mM glucose with CIN, followed by CAP failure, whereas with 30 mM glucose and CIN, the CAP recovery was sustained (Figure 6C). A simultaneous recording with a lactate sensor placed at the nerve boundary [4,14], which records release of lactate, presumably from Schwann cells, temporally aligns CAP rescue with significant lactate release. These data indicate that CAP rescue requires functional Schwann cells and lactate uptake into axons. The Schwann cells may be also required to buffer the elevations in interstitial K^+^ that accompany high-frequency firing in the sciatic nerve [15].

## 4. Methods

### 4.1. Ethical Approval

All experiments were approved by the University of Nottingham Animal Care and Ethics Committee; were carried out in accordance with the Animals (Scientific Procedures) Act 1986 under appropriate authority of establishment, project, and personal licences; and conform to the principles and regulations described in the Editorial by Grundy [16]. Experiments were performed on male CD-1 mice (weight 28–35 g, corresponding to 30–45 days of age) purchased from Charles River Laboratories (Margate, UK). Mice were group housed with ad libitum access to food and water and maintained at 22–23 °C on a 12 h–12 h light–dark cycle. Mice were killed by Schedule 1 cervical dislocation; death was confirmed by permanent cessation of the circulation. A total of 76 mice were used, providing data from 115 nerves. Power calculations associated with recordings of two nerves from the same animal in separate chambers, and details on appropriate sample sizes for known effect sizes, are available [17].

### 4.2. Electrophysiology

The nerves were allowed to equilibrate in the interface superfusion chamber (Medical Systems Corp, Greenvale, NY, USA) for 10–15 min prior to recording. The nerves were maintained at 37 °C and superfused with aCSF containing (in mmol^−l^): NaCl 126, KCl 3.0, CaCl_2_ 2.0, MgCl_2_ 2.0, NaH_2_PO_4_ 1.2, NaHCO_3_ 26 at a rate of about 2 mL min^−1^. Control aCSF contained 10 mM glucose. The substrate in the aCSF can be omitted completely (substrate-free), or replaced with another, in order to study aglycaemia. The chamber was continuously aerated by a humidified gas mixture of 95% O_2_–5% CO_2_. The electrophysiological recording process has been described in detail [9], but briefly, suction electrodes backfilled with substrate-free aCSF, to preclude creation of a substrate-rich reservoir, were used for stimulation and recording. Nerves were gently introduced into the suction electrodes, which were manufactured as previously described [18]. The recordings of the stimulus-evoked A fibre CAPs or optic nerves were controlled by proprietary software. The two sciatic nerves from each mouse were recorded in separate chambers to reduce animal use while maintaining statistical power [17]. The nerves were stimulated using a Grass S88 stimulator connected to an SIU5 isolation unit. The signal was amplified up to ×1000 in AC mode by a Stanford Research Systems Preamplifier (SR560, Stanford Research Systems, Sunnyvale, CA, USA), filtered at 30 kHz, and acquired at 20 kHz (Clampex 9.2, Molecular Devices, Wokingham, UK). High-frequency stimulus (HFS) was defined as 100 Hz stimulus.

### 4.3. Lactate Biosensors

Lactate biosensors were purchased from Sarissa Biomedical (Coventry, UK). The lactate biosensors (25 μm in diameter and 500 μm in length) were pressed against the sciatic nerve, to record (lactate) release from the nerve as previously described [9]. Experimental recordings began after an equilibration period of up to an hour. At the beginning and end of all experiments, lactate biosensors were calibrated using lactate concentrations of 10 μM and 100 μM. Results were considered valid only if the before and after calibrations deviated by no more than 10%.

### 4.4. Statistical Analysis

Descriptive statistics are expressed as the mean ± standard deviation, where *n* refers to the number of nerves [17]. The latencies to CAP failure and area of the CAP were calculated as previously described [9]. One-way ANOVA with Sidak post-tests was used to determine the significance of selected inter-group differences. All data were analysed with GraphPad Prism 7 (La Jolla, CA 92037, USA). Statistical significance in Figure 1, Figure 2, Figure 3, Figure 4, Figure 6 and Figure 7: *p* < 0.05 = *; *p* < 0.01 = **; *p* < 0.005 = ***; *p* < 0.001 = ****.

## 5. Conclusions

We have described electrophysiological recordings of the stimulus-evoked A fibre CAP that illuminate substrate use in the sciatic nerve during baseline and increased tissue energy demand. The balance between tissue energy demand, which can be varied according to imposed stimulus frequency, and substrate supply, which can be varied in the aCSF, reveals that CAP conduction is lost when stimulus frequency is increased but can be restored if the substrate concentration is increased. In the absence of exogenously applied substrate, Schwann cell glycogen is metabolised to lactate, which supports axon conduction. Increasing tissue energy demand leads to CAP failure in the presence of normoglycaemic glucose, but hyperglycaemic glucose fully maintains the CAP and promotes glycogen storage under equivalent conditions. In the absence of glycogen, CAP conduction can be rescued by exogenously applied substrate alone but requires the presence of functional Schwann cells. These findings contribute to the understanding of the pathophysiology of metabolic disorders, including diabetes, to which peripheral nerves are susceptible.

## Figures and Tables

**Figure 1 metabolites-12-00505-f001:**
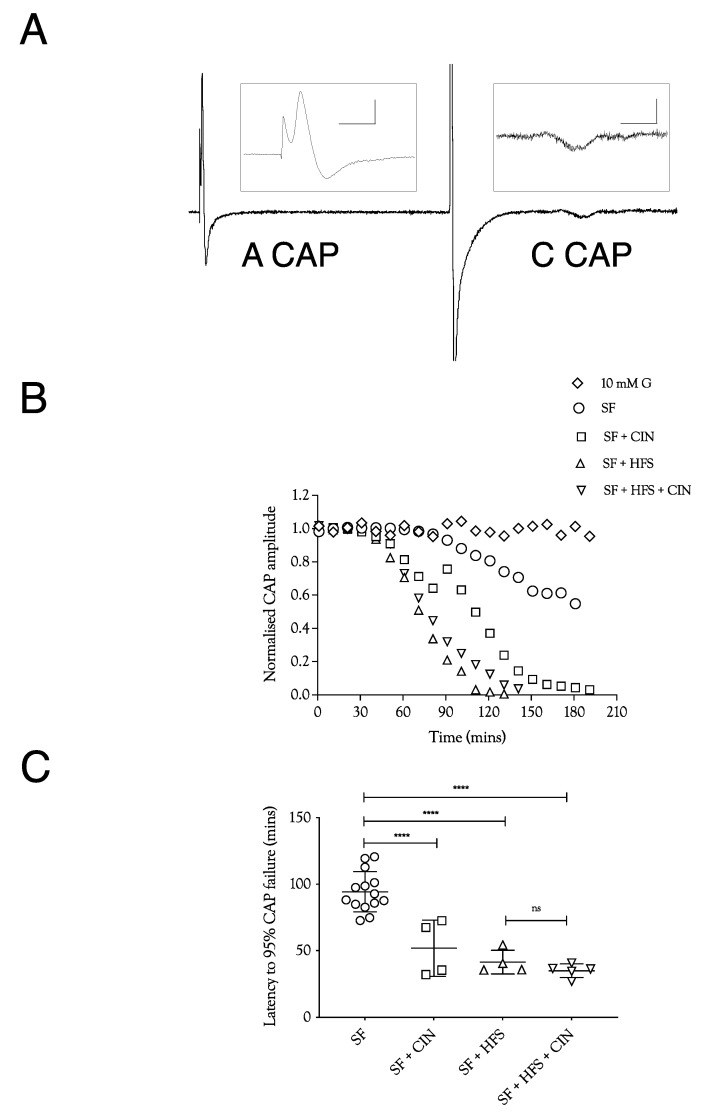
A fibre CAP conduction in substrate-free (SF) aCSF. (**A**). Specimen records of ex vivo sciatic nerves stimulated electrically, showing the fast-conducting A fibre CAP and the slow-conducting C fibre CAP. Stimulation was adjusted to elicit a maximal A fibre CAP amplitude. The behaviour of the low-amplitude C fibre CAP was not evaluated further. Scale bars are 5 ms and 2 mV for the A fibre CAP and 2 ms and 10 μV for the C fibre CAP. (**B**). Perfusing sciatic nerves with substrate-free (SF) aCSF resulted in a slow reduction in A fibre CAP amplitude. The speed of failure was accelerated by adding 200 μM cinnamate (CIN), high-frequency stimulation (HFS), or both. (**C**). In substrate-free conditions, the A fibre CAP was maintained for 94.3 ± 15.1 min (*n* = 14), which fell to 51.9 ± 21 min (*n* = 4) on addition of CIN, 41.5 ± 8.8 min, (*n* = 4) on exposure to high-frequency stimulus, and 35.1 ± 5 min (*n* = 5) with both CIN and high-frequency stimulus. (G = glucose, L = lactate, HFS = high-frequency stimulus, CIN = cinnemate: applies to all figures; **** = *p* ≤ 0.0001, ns = not significant).

**Figure 2 metabolites-12-00505-f002:**
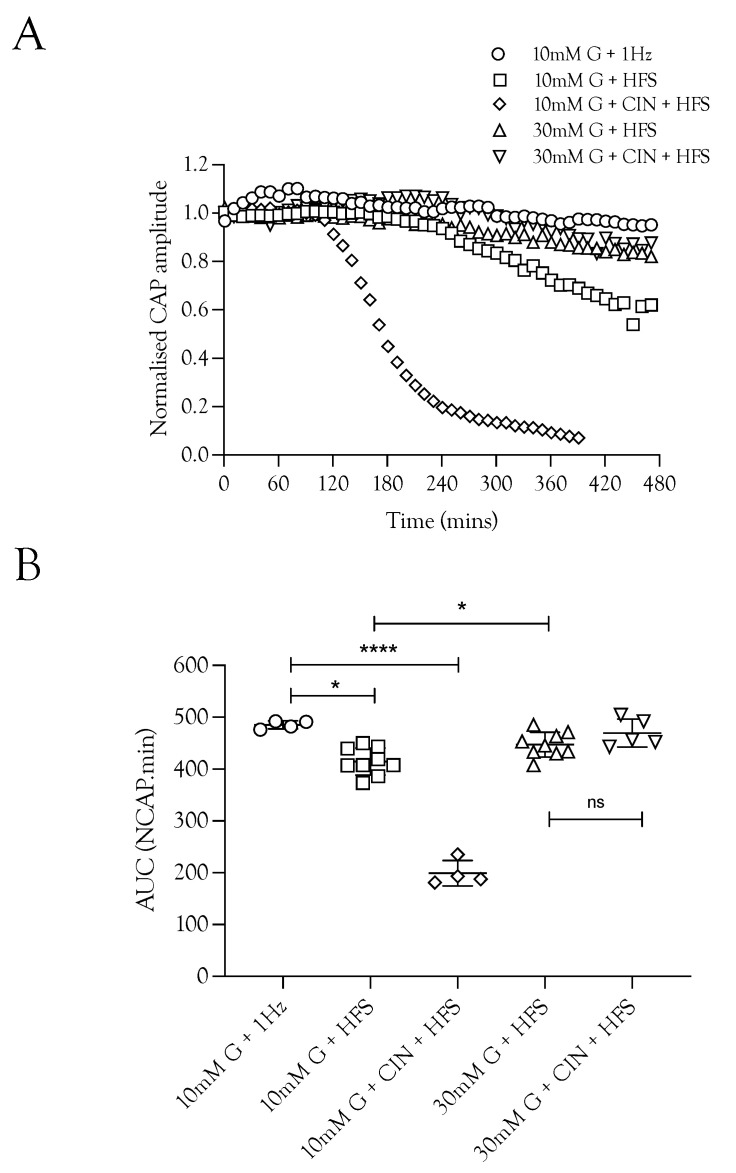
The increased energy demand placed on the sciatic nerve by HFS was met by increased energy substrate supply. (**A**). The A fibre CAP was fully supported in 10 mM glucose but decreased when exposed to HFS. However, increasing the glucose from 10 mM to 30 mM in nerves exposed to HFS increased the maintenance of the CAP. The addition of 200 μM CIN during 10 mM glucose and HFS resulted in accelerated failure of conduction with both reduced latency to failure onset (approximately 2 h compared to 4 h) and increased rate of failure. (**B**). Increasing the stimulus frequency under normoglycaemic conditions (10 mM glucose) reduced the maintenance of conduction from 485.4 ± 7.6 NCAP.mins (*n* = 3) to 414.9 ± 26.0 NCAP.mins (*n* = 9). The CAP was increased when the 10 mM glucose was replaced with 30 mM glucose (447.2 ± 23.8 NCAP.mins, *n* = 9), but addition of CIN had no effect on the CAP area (469.9 ± 27.0 NCAP.mins, *n* = 5). The addition of CIN reduced the latency to CAP failure in nerves perfused with 10 mM glucose and exposed to HFS (199.3 ± 24.5 NCAP.mins, *n* = 4) (**** = *p* ≤ 0.0001, * = *p* ≤ 0.05, ns = not significant).

**Figure 3 metabolites-12-00505-f003:**
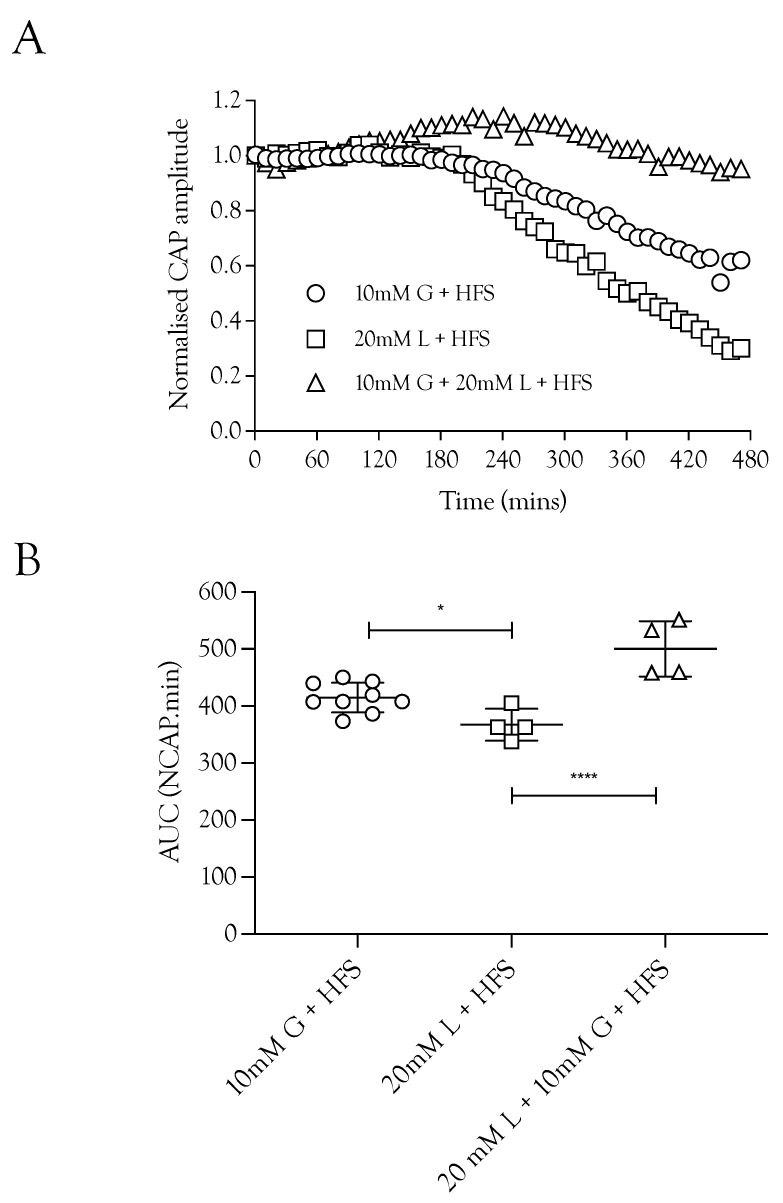
Lactate acts as an energy source for A fibres during HFS. (**A**) Substitution of 10 mM glucose with 20 mM lactate during HFS significantly reduced the CAP area. The combination of 10 mM glucose and 20 mM lactate enabled full maintenance of conduction. (**B**) The A fibre CAP was partially maintained in 10 mM glucose and HFS (414.9 ± 26.0 NCAP.mins, *n* = 9). Replacement of glucose with 20 mM lactate decreased CAP support (367.2 ± 28 NCAP.mins, *n* = 4), but combination of 10 mM glucose and 20 mM lactate increased the maintenance of conduction compared to 10 mM glucose alone during HFS (500.3 ± 48.7 NCAP.mins, *n* = 4) (**** = *p* ≤ 0.0001, * = *p* ≤ 0.05).

**Figure 4 metabolites-12-00505-f004:**
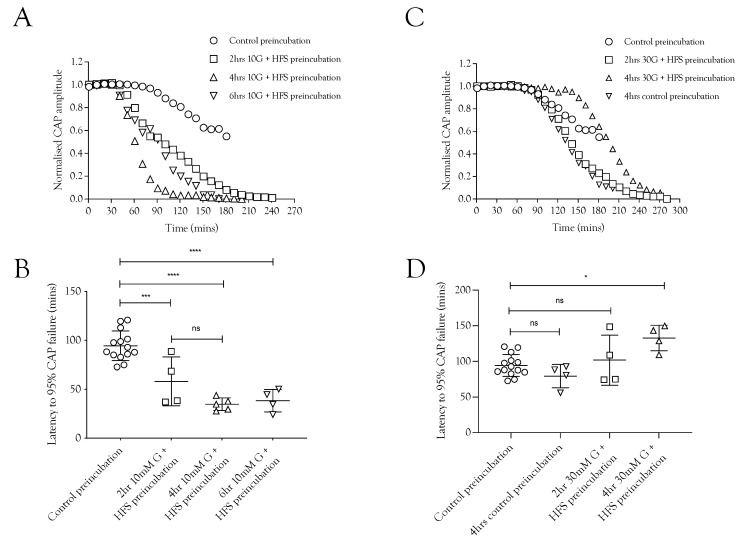
Glycogen is metabolised during normoglycemia (10 mM glucose) and HFS but synthesised during increased glucose supply and HFS. (**A**) Conduction failure during removal of exogenous energy substrate supply was accelerated as a result of exposure to HFS when supplied with 10 mM glucose compared to baseline conditions. (**B**) The latency to CAP failure decreased from 94.3 ± 15.1 min (*n* =14) to 58 ± 24.9 min (*n* = 4) after 2 h pre-incubation in 10 mM glucose with HFS, an effect that was not significantly increased when the duration was increased to 4 h (34.7 ± 6.5 min, *n* = 5) or 6 h (38.3 ± 11.5 min, *n* = 4). (**C**) Pre-incubating the nerve in 10 mM glucose and HFS for increasing durations reduced the latency to A fibre CAP failure during subsequent substrate-free conditions. (**D**) Latency to A fibre CAP failure during substrate-free conditions was increased from 79.5 ± 16.3 min (*n* = 4) to 94.3 ± 15.1 min (*n* = 14) when the pre-incubation duration was increased in 30 mM glucose to 4 h. Greater increases in the latency to CAP failure occurred as a result of increasing the glucose to 30 mM, where latency increased to 101.7 ± 35.1 min after 2 h (*n* = 4) and 132.8 ± 18.0 min after 4 h (*n* = 4) (**** = *p* ≤ 0.0001, *** = *p* ≤ 0.001, * = *p* ≤ 0.05, ns = not significant).

**Figure 5 metabolites-12-00505-f005:**
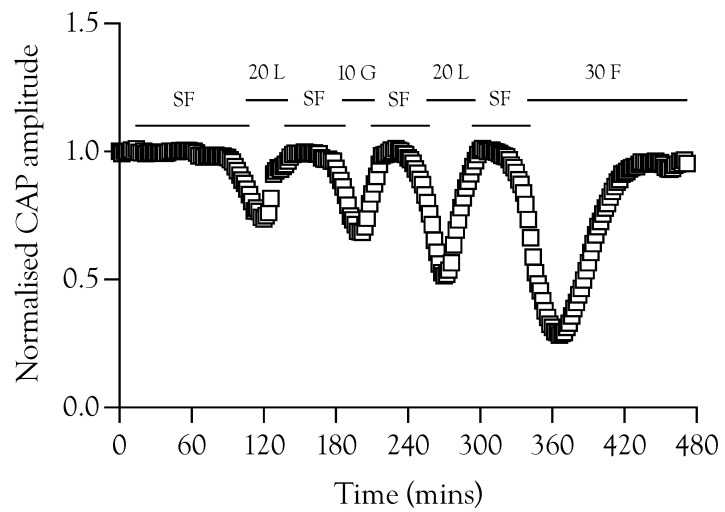
The ability of substrates to recover failing A fibre CAP. A sciatic nerve was repeatedly exposed to substrate-free aCSF (SF) in order to promote CAP failure. The CAP was fully restored on introduction of aCSF containing 10 mM glucose (10 G), 20 mM lactate (20 L), or 30 mM fructose (30 F).

**Figure 6 metabolites-12-00505-f006:**
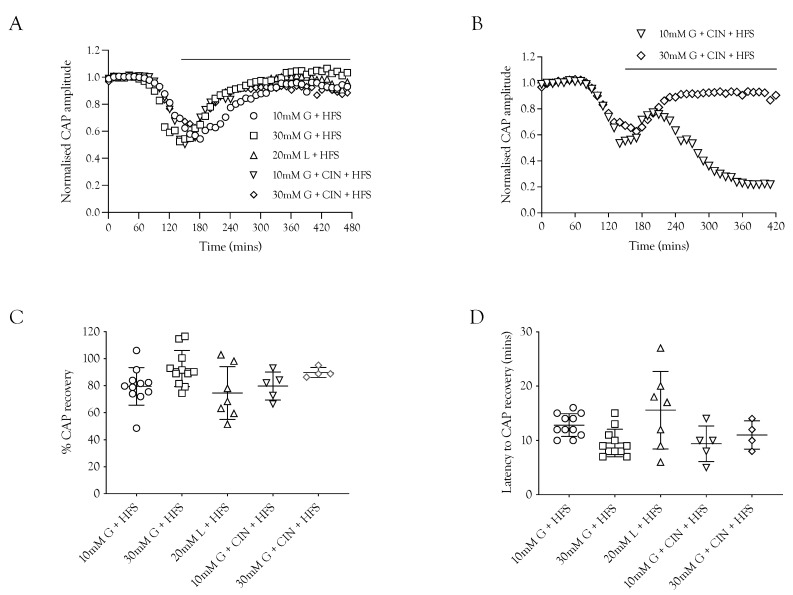
The A fibre CAP can be restored in the absence of glycogen. (**A**) After exposure to substrate-free conditions resulting in a drop in amplitude, the A fibre CAP was restored by glucose and lactate during HFS (indicated by the horizontal bar). (**B**) The recovery was sustained in 30 mM glucose and CIN but was only temporarily rescued in 10 mM glucose and CIN. (**C**) The A fibre CAP was restored to a similar level under all 5 conditions of HFS (79.5 ± 14.0%, *n* = 11 (10 mM glucose) vs. 92.7 ± 13.4%, *n* = 11 (30 mM glucose), 74.5 ± 19.6%, *n* = 7 (20 mM lactate), 79.7 ± 10.4%, *n* = 5 (10 mM glucose and 200 μM CIN), and 89.8 ± 3.7%, *n* = 4 (30 mM glucose and 200 μM CIN). (**D**) The time in which the A fibre CAP starts to recover during HFS after the onset of energy substrate supply was equal amongst all 5 conditions (12.8 ± 2.1 min, n = 11 (10 mM glucose) vs. 9.5 ± 2.5 min, *n* = 11 (30 mM glucose), 15.6 ± 7.1 min, *n* = 7 (20 mM lactate), 9.4 ± 3.3 min, *n* = 5 (10 mM glucose and 200 μM CIN), and 11.0 ± 2.6 min, *n* = 4 (30 mM glucose and 200 μM CIN).

**Figure 7 metabolites-12-00505-f007:**
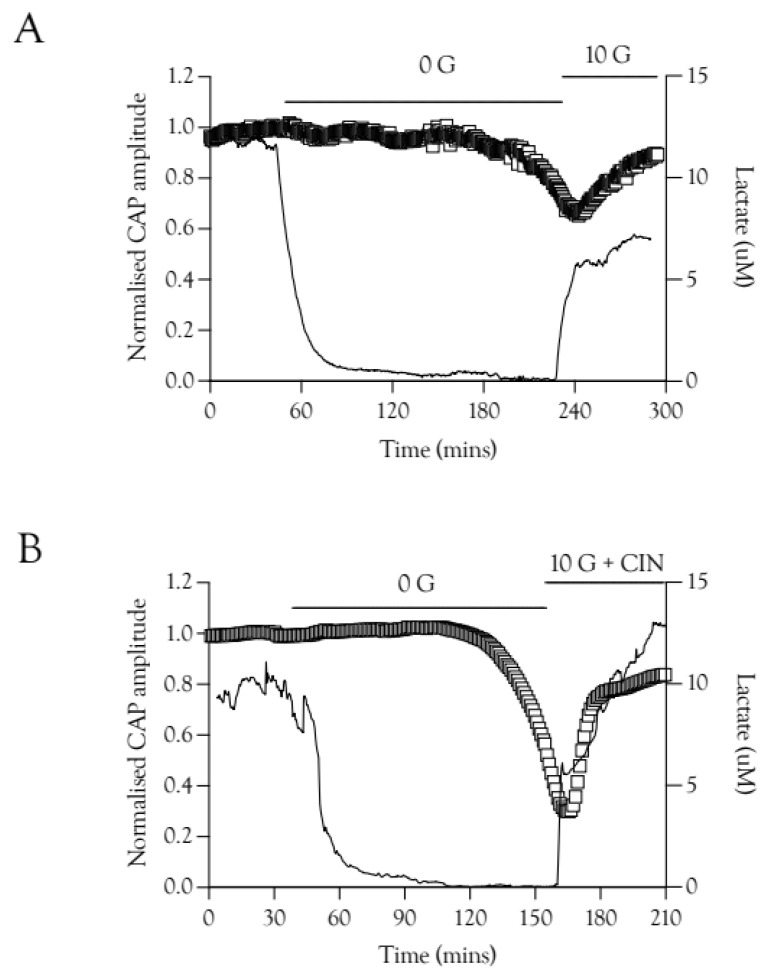
CAP recovery is preceded by elevations in extracellular lactate [lac]_o_. (**A**). Recovery of the A fibre CAP is delayed compared to recovery of [lac]_o_ in 10 mM glucose. There is a temporal correlation between the onset of CAP recovery and the peak [lac]_o_ level reached after introduction of 10 mM glucose, *n* = 3. (**B**). A similar effect was seen in recovery with 10 mM glucose and CIN, where the CAP recovers only after an initial rapid increase in [lac]_o_, *n* = 3.

## Data Availability

The data presented in this study are available on request from the corresponding author. The data are not publicly available due to limited funds.

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
