# Peer review of "Energy Metabolism in Mouse Sciatic Nerve A Fibres during Increased Energy Demand"

_metabolites, 2022, doi:10.3390/metabo12060505_

Round 1

Reviewer 1 Report

The manuscript entitled "Energy metabolism in mouse sciatic nerve A fibres during increased energy demand" by Rich et al. presents a convincing line of evidence that neurolemmocytes-derived lactate supports axonal conduction in the absence of exogenous energy substrates. The study is the yet another report which supports the astrocyte-neuron lactate shuttle hypothesis, whatmore, it fully justifies the extension of the hypothesis far beyound the physiology of astrocytes. The results put an accent on an importance of all glial cells in the functioning of nervous system, in the context of this study particularly of peripheral nervous system.

I have neighter methodological nor meritoric concerns regarding the study.

Please, revise the figures and the figures legends. Some panels are not correctly described in the legend. Also, please unify the glucose description in the figures (sometimes it is "glucose" and sometimes it is just "G").

I fully recommend the manuscript to be published in Metabolites. Nice work!

Reviewer 2 Report

In the manuscript "Energy Metabolism in the Sciatic Nerve Fibers of Mouse A During Increased Energy Demand" both the introduction and the discussion are clear and consistent with the results.

The results are described in an exhaustive way but the images in particular the writings are grainy. A note: the authors do not describe the scientific context in which these findings make a scientific contribution.

I deduced that the aim of the manuscript falls into a field of study on damage to peripheral nerves (sciatic nerve) in diabetic subjects, if this were to be specified more clearly.
